# Serum syndecan-1 concentration in hospitalized patients with heart failure may predict readmission-free survival

**Yuichiro Kitagawa[1]©, Itta Kawamura[2]©, Keiko Suzuki[3]©, Hideshi Okada[1]©\*, Takuma Ishihara[4], Hiroyuki Tomita[5], Kodai Suzuki[1], Chihiro Takada[1], So Sampei[1], Soichiro Kano[1], Kohei Kondo[1], Hirotaka Asano[1], Yugo Wakayama[1], Ryo Kamidani[1], Yuki Kawasaki[1], Hirotsugu Fukuda[1], Ayane Nishio[1], Takahito Miyake[1], Tetsuya Fukuta[1], Ryu Yasuda[6], Hideaki Oiwa[1], Yoshinori Kakino[1], Nagisa Miyazaki[7], Takatomo Watanabe[8], Takahiro Yoshida[1], Tomoaki Doi[1], Akio Suzuki[3], Shozo Yoshida[6], Hitoshi Matsuo[2], Shinji Ogura[1]**

1 Department of Emergency and Disaster Medicine, Gifu University Graduate School of Medicine, Gifu, Japan, 2 Gifu Heart Center, Gifu, Japan, 3 Department of Pharmacy, Gifu University Hospital, Gifu, Japan, 4 Innovative and Clinical Research Promotion Center, Gifu University Hospital, Gifu, Japan, 5 Department of Tumor Pathology, Gifu University Graduate School of Medicine, Gifu, Japan, 6 Department of Abuse Prevention Emergency Medicine, Gifu University Graduate School of Medicine, Gifu, Japan, 7 Department of Internal Medicine, Asahi University School of Dentistry, Mizuho, Japan, 8 Department of Clinical Laboratory, Gifu University Hospital, Gifu, Japan

© These authors contributed equally to this work.
\* hideshi@gifu-u.ac.jp

**Data Availability Statement:** Data are available from the medical ethics committee of Gifu University Graduate School of Medicine (rinri@gifu-u.ac.jp) for researchers who meet the

## Abstract

Syndecan-1 is found in the endothelial glycocalyx and is released into the bloodstream during stressed conditions, including severe diseases such as acute kidney injury, chronic kidney disease, and cardiovascular disease. This study investigated the prognostic value of serum syndecan-1 concentration in patients with heart failure upon admission. Serum syndecan-1 concentration was analyzed in 152 patients who were hospitalized for worsening heart failure from September 2017 to June 2018. The primary outcome of the study was readmission-free survival, defined as the time from the first admission to readmission for worsened heart failure or death from any cause, which was assessed at 30 months after discharge from the hospital. The secondary outcome of the study was survival time. Blood samples and echocardiogram data were analyzed. Univariate and multivariable time-dependent Cox regression analyses adjusted for age, creatinine levels, and use of antibiotics were conducted. The serum syndecan-1 concentration was significantly associated with readmission-free survival. Subsequently, the syndecan-1 concentration may have gradually decreased with treatment. The administration of human atrial natriuretic peptide and antibiotics may have modified the relationship between readmission-free survival and serum syndecan-1 concentration (p = 0.01 and 0.008, respectively). Serum syndecan-1 concentrations, which may indicate injury to the endothelial glycocalyx, predict readmission-free survival in patients with heart failure.

criteria for access to confidential data, because data contain potentially identifying information.

**Funding:** Japan Society for the Promotion of Science 20K17888 Dr. Ryu Yasuda Japan Society for the Promotion of Science 20K17857 Dr. Yuichiro Kitagawa Japan Society for the Promotion of Science 20K17856 Dr. Yoshinori Kakino Japan Society for the Promotion of Science 19H03756 Dr. Hideshi Okada Japan Society for the Promotion of Science 19K09410 Dr. Tomoaki Doi Japan Society for the Promotion of Science 19K18347 Dr. Tetsuya Fukuta Japan Society for the Promotion of Science 18K08914 Dr. Kodai Suzuki Japan Society for the Promotion of Science 18K08884 Dr. Takahiro Yoshida Japan Society for the Promotion of Science 18K16511 Prof. Shozo Yoshida Japan Society for the Promotion of Science 17K11569 Prof. Shinji Ogura.

**Competing interests:** The authors declare no conflict of interest relevant to the content of this manuscript.

## Introduction

Decompensated heart failure is defined as a clinical syndrome in which a structural or functional change in the heart leads to its inability to eject and/or accommodate blood within the physiological pressure levels [1]. As the pathophysiology of decompensated heart failure is complicated and heterogeneous [2], an accurate assessment of severity remains a significant challenge. It is important to predict the prognosis of patients with heart failure to provide precise medical care.

Currently, risk stratification of patients with non-ischemic heart failure is commonly based on clinical parameters, such as the New York Heart Association class; echocardiographic parameters, such as left ventricular ejection fraction (LVEF); and blood markers, such as the level of a 76-amino acid N-terminal fragment in the prohormone called B-type natriuretic peptide and brain natriuretic peptide [3, 4]. However, the use of plasma brain natriuretic peptide, while very valuable in differentiating heart failure from noncardiac causes of acute dyspnea [5], is debatable as a prognostic marker [6–8]. Currently, there is no class I indication in the guidelines for the use of specific biomarkers for prognostic purposes [9, 10].

Pleural effusions are commonly recognized in patients with congestive heart failure. Transudative pleural effusion is caused by various factors such as increasing capillary hydrostatic pressure or decreasing colloid oncotic pressure.

The sugar–protein glycocalyx covers the surface of a healthy vascular endothelium [11–13] and maintains vascular homeostasis, such as by regulating an oncotic gradient across the endothelial barrier [14–18]. The endothelial glycocalyx may get injured under abnormal conditions such as sepsis, diabetes, and hypertension, thereby increasing vascular permeability [15, 19, 20]. The removal of glycocalyx has been known to cause a dramatic increase in hydraulic permeability [21]. Furthermore, previous studies have found that rapid plasma volume expansion injured the endothelial glycocalyx [22].

The endothelial glycocalyx consists of cell-bound proteoglycans, glycosaminoglycan side chains, and sialoproteins [23–25]. Proteoglycans are composed of a core protein, such as a syndecan family protein, to which glycosaminoglycan is linked. Syndecan-1 is the core protein in heparan sulfate proteoglycan, which is also found in the endothelial glycocalyx. Syndecan-1 detaches from the endothelium upon glycocalyx injury, thereby increasing its serum concentration [26]. Therefore, serum syndecan-1 has been used as a marker of endothelial injury in recent clinical studies [27, 28]. Because the pulmonary capillary glycocalyx is thinner than other organs [29], it may be easily injured by excess fluid. Therefore, transudative pleural effusion may occur due to heart failure. Moreover, the concentration of serum syndecan-1 has been found to be associated with clinical outcomes in patients with heart failure and preserved ejection fraction [30]. Additionally, it has been suggested that syndecan-1 is associated with left ventricular hypertrophy in heart failure with preserved ejection fraction [31].

Therefore, in this study, we investigated the prognostic value of serum syndecan-1 concentration upon admission in patients with heart failure.

## Materials and methods

### Patient profile

Patients who were hospitalized at Gifu Heart Center due to worsening heart failure between September 2017 and June 2018 were included in this study. Those who were <20 years old and/or had acute coronary syndrome, malignant tumor, liver cirrhosis, collagen disease, and hemodialysis were excluded from the analysis. Overall, 152 patients were enrolled in this study, and 784 samples were obtained.

## Data collection and study design

Demographic and clinical data, including medical and medication histories, were collected. The following data were included in the time-dependent Cox regression model: age, sex, serum albumin, aspartate aminotransferase (AST), alanine aminotransferase (ALT), creatine kinase (CK), triglyceride, total cholesterol, high-density lipoprotein cholesterol, low-density lipoprotein cholesterol, blood urea nitrogen (BUN), creatinine (Cre) concentration, sodium, potassium, chlorine, C-reactive protein, estimated glomerular filtration rate (eGFR), white blood cell number, hemoglobin (Hb) concentration, hematocrit (Ht), hemoglobin A1c, left ventricular diastolic diameter (LVDd), left ventricular systolic diameter (LVDs), interventricular septum thickness (IVST), posterior wall thickness (PWT), LVEF, left atrial dimension (LAD), left atrial volume index (LAVI), and the diameter of the inferior vena cava (IVC).

Echocardiography was performed at the time of admission. Blood samples were obtained when the physician deemed it necessary, and they were collected into the sample collection tube with a serum separating medium. The serum was collected and preserved in a freezer at −80 ˚C. Serum syndecan-1 concentrations were measured using enzyme-linked immunosorbent assay (950.640.192, Diaclone, Besancon, Cedex, France). Sequential samples from the same patient were examined in the same enzyme-linked immunosorbent assay plate to avoid inter-assay variability. The primary outcome of the study was readmission-free survival, defined as the time from the first admission to readmission for worsened heart failure or death from any cause, which was assessed at 30 months after discharge from the hospital. The secondary outcome of the study was survival time.

## Statistical analyses

Descriptive statistics are presented as frequencies and percentages for categorical variables and as medians with interquartile ranges for continuous variables. For the time-to-event analyses, enrolled patients were monitored from the study initiation until December 2019. Cumulative readmission-free survival rates, stratified by median of serum syndecan-1 concentration at baseline, were obtained with Kaplan–Meier estimation. To assess the effect of serum syndecan-1 concentration on readmission-free survival time and mortality, time-dependent Cox proportional hazards multivariable regression analyses were performed. Covariates in the model, including age [32, 33], Cre [33–36], and antibiotic use, indicating the presence of an infectious disease [18, 27, 28, 37], were selected a priori for their clinical relevance. Serum syndecan-1 concentration, Cre, and antibiotic use were treated as time-dependent covariates in the model. Additionally, serum syndecan-1 concentration was modeled using a restricted cubic spline to allow for nonlinear associations.

Furthermore, to assess whether the effects of serum syndecan-1 concentration were modified by blood and echocardiogram parameters, a cross-product term between serum syndecan-1 concentration and variables of parameters was included in the time-dependent Cox regression (i.e., interaction) analysis. A two-sided significance level of 0.05 was used for all statistical inferences. All data analyses were performed using the R statistical software, version 3.6.2 (R Foundation for Statistical Computing, Vienna, Austria. URL https://www.R-project. org/.).

## Ethics statement

The investigation conformed with the principles outlined in the Declaration of Helsinki [38]. Ethics approval was obtained from the medical ethics committee of Gifu University Graduate School of Medicine, Gifu, Japan (record no.: 29–216), and Gifu Heart Center, Gifu, Japan

(record no.: 2017022). All patients provided written informed consent for their participation in the study as well as for the publication of this report.

## Results

### Patient characteristics

The demographic data of patients are presented in Table 1. Altogether, 152 patients (94 men and 58 women) with a median age of 76 (range, 68–85) years were enrolled in the study. These patients had basal heart disease, and all patients received medications, including various combinations. Differences in syndecan-1 concentrations among patients with different origins of heart failure are shown in S1 Table and those among patients with heart failure due to different medications are shown in S2 Table.

### Serum syndecan-1 and readmission-free survival in patients with heart failure

The Kaplan–Meier curve showed that the median readmission-free survival time, stratified by median of serum syndecan-1 concentration at baseline, was 331 (<33.48 ng/mL) and 239 (≥33.48 ng/mL) days (Fig 1). The median readmission-free survival time of all patients was 288 days (event number = 67, 95% confidence interval [CI]: 227–356). Univariate and multi-variable time-dependent Cox regression analyses adjusted for age, Cre, and antibiotic use, excluding the effect of infection, were conducted (Table 2 and Fig 2). We evaluated the effect on readmission-free survival using hazard ratios (HRs) when the value of serum syndecan-1 concentration changed from median among all data including repeated measurements (37.04 ng/mL) to 50 ng/mL, 100 ng/mL, 200 ng/mL, 300 ng/mL, and 400 ng/mL. With a change in serum syndecan-1 concentration from 37.04 ng/mL to 300 ng/mL, the unadjusted time-dependent Cox proportional hazards model revealed an association between serum syndecan-1 concentration and readmission-free survival (HR, 1.687; 95% CI, 1.1–2.585; p = 0.016). The analysis adjusted for covariates also showed that the serum syndecan-1 concentration was associated with readmission-free survival (HR, 1.865; 95% CI, 1.169–2.974; p = 0.009). The increase in serum syndecan-1 concentration from 37.04 ng/mL to 300 ng/mL was associated with an approximately 1.9-fold increase in the risk of readmission-free survival after adjusting for confounders.

The average concentration of syndecan-1 in serum was 72.4 ng/mL (95% CI, 37.5–107.2) upon admission, which peaked at 101.21 ng/mL (95% CI, 69.77–132.66 ng/mL) at 2.77 days after admission. Subsequently, the syndecan-1 concentration decreased gradually, probably as a result of treatment (Fig 3).

Table 3 shows the p-values for interaction of the parameters modifying the relationship between readmission-free survival and serum syndecan-1 concentration. The administration of **human atrial natriuretic peptide** and antibiotics appeared to modify the relationship between readmission-free survival and serum syndecan-1 concentration (p for interaction = 0.01 and 0.008, respectively; Fig 4). Moreover, AST, ALT, CK, BUN, Hb, Ht, PWT, and LAVI appeared to modify the relationship due to heart failure (p for interaction, 0.021, 0.001, <0.001, 0.003, 0.014, 0.004, 0.045, and 0.031, respectively; Fig 4 and S1 Fig). Likewise, BUN/Cre appeared to modify the relationship due to heart failure (S3 Table). Conversely, the LVEF did not appear to modify the relationship between readmission-free survival and serum syndecan-1 concentration (p for interaction = 0.418).

**Table 1. Patients' demographics.**

| Characteristics | N = 152 |
| --- | --- |
| Age (years), median (IQR) | 76 (68–85) |
| Sex (Female/Male), n (%) | 58 (38.2) / 94 (61.8) |
| Follow-up time per patient (days), median (IQR) | 23 (11–230) |
| Number of measurements per patient, median (IQR) | 4 (3–6) |
| Death, n (%) | 21 (13.8) |
| Readmission, n (%) | 46 (30.3) |
| Basal heart disease, n (%) | |
| Hypertensive heart disease | 26 (17.1) |
| Ischemic heart disease (post-PCI) | 21 (13.8) |
| Ischemic heart disease (post-CABG) | 12 (8.9) |
| Ischemic heart disease (conservative treatment) | 7 (4.6) |
| Arrhythmia (tachycardia) | 20 (13.2) |
| Arrhythmia (bradycardia) | 1 (0.7) |
| Dilated cardiomyopathy | 17 (11.2) |
| Hypertrophic cardiomyopathy | 5 (3.3) |
| Other cardiomyopathy | 6 (4.0) |
| Aortic valve stenosis (post-operation) | 6 (4.0) |
| Aortic valve stenosis (conservative treatment) | 3 (2.0) |
| Aortic valve insufficiency (post-operation) | 1 (0.7) |
| Aortic valve insufficiency (conservative treatment) | 5 (3.3) |
| Mitral valve insufficiency (post-operation) | 3 (2.0) |
| Mitral valve insufficiency (conservative treatment) | 8 (5.3) |
| Tricuspid valve insufficiency (conservative treatment) | 3 (2.0) |
| Congestive disease | 2 (1.3) |
| Other | 6 (3.9) |
| Medication, n (%) | |
| Beta blocker | 123 (80.9) |
| ACE inhibitor/ARB | 100 (65.8) |
| Statin | 61 (40.1) |
| Antiplatelet | 65 (42.8) |
| Anticoagulant | 76 (50.0) |
| Loop diuretic | 136 (89.5) |
| Spironolactone | 100 (65.8) |
| Tolvaptan | 60 (39.5) |
| Catecholamine | 23 (15.1) |
| Human atrial natriuretic peptide | 96 (63.2) |
| Antibiotic | 43 (28.3) |
| Echocardiographic characteristics, median (IQR) | |
| LVDd (mm) | 53.2 (46.0–58.1) |
| LVDs (mm) | 41.4 (30.4–51.0) |
| IVST (mm) | 9.0 (8.3–9.7) |
| PWT (mm) | 9.3 (8.4–10.2) |
| LVEF (%) | 40.5 (25.6–59.6) |
| LAD (mm) | 46.7 (43.7–52.0) |
| LAVI | 60.2 (47.4–67.7) |
| IVC (mm) | 8.6 (6.4–12.1) |

IQR: interquartile range, PCI: percutaneous coronary intervention, CABG: coronary artery bypass grafting, ACE: angiotensin-converting enzyme, ARB: angiotensin II receptor blocker, LVDd: left ventricular diastolic diameter, LVDs: left ventricular systolic diameter, IVST: interventricular septum thickness, PWT: posterior wall thickness, LVEF: left ventricular ejection fraction, LAD: left atrial dimension, LAVI: left atrial volume index, IVC: diameter of the inferior vena cava.

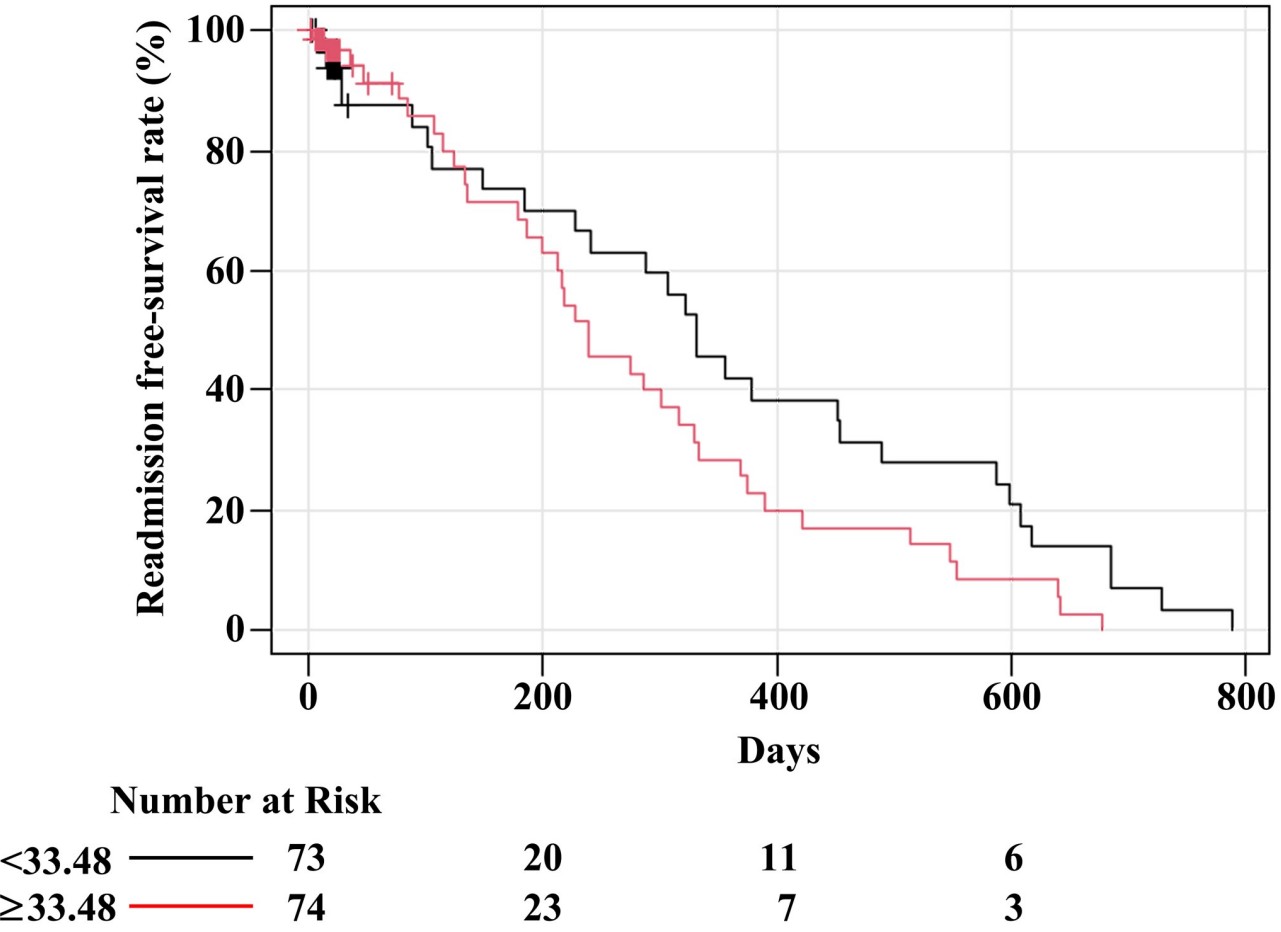

**Fig 1. Kaplan–Meier curve of groups stratified by median of serum syndecan-1 concentration at baseline.** The median readmission-free survival times of groups stratified by median of serum syndecan-1 concentration are 331 (<33.48 ng/mL) and 239 (≥33.48 ng/mL) days. The number of events is 30 (<33.48 ng/mL) and 36 (≥33.48 ng/mL).

## Discussion

The present study revealed that serum syndecan-1 concentration: 1) was higher in patients with heart failure than in healthy subjects from previous reports [33], 2) responded promptly to treatment against heart failure, and 3) might predict readmission-free survival in patients with heart failure.

### Serum syndecan-1 concentration in heart failure

A previous study has also reported that syndecan-1 concentration was associated with clinical outcomes in patients with heart failure and with preserved ejection fraction [30]. The serum syndecan-1 concentration in patients with heart failure in this study was 72.4 ng/mL at admission, which peaked to 101.21 ng/mL on day 3 of admission. Although values are higher than those reported in our previous studies, where the serum syndecan-1 concentration was 13.7–27.3 ng/mL in healthy individuals [33], the origin of syndecan-1 remains controversial.

**Table 2. The relationship between serum syndecan-1 concentration and readmission /death.**

| | Univariate analysis | | | Multivariable analysis | | |
|---|---|---|---|---|---|---|
| | HR | 95% CI | P value | HR | 95% CI | P value |
| Syndecan-1 (ng/mL) 37.04 (median)–50 | 0.981 | 0.85–1.133 | 0.798 | 1 | 0.873–1.145 | >0.999 |
| Syndecan-1 (ng/mL) 37.04 (median)–100 | 0.994 | 0.631–1.567 | 0.981 | 1.061 | 0.691–1.629 | 0.787 |
| Syndecan-1 (ng/mL) 37.04 (median)–200 | 1.271 | 0.796–2.03 | 0.316 | 1.388 | 0.875–2.204 | 0.164 |
| Syndecan-1 (ng/mL) 37.04 (median)–300 | 1.687 | 1.1–2.585 | 0.016 | 1.865 | 1.169–2.974 | 0.009 |
| Syndecan-1 (ng/mL) 37.04 (median)–400 | 2.238 | 1.423–3.521 | <0.001 | 2.504 | 1.457–4.305 | 0.001 |
| Age (years) | 1.015 | 0.994–1.038 | 0.168 | 1.011 | 0.989–1.033 | 0.336 |
| Sex (Male) | 0.673 | 0.417–1.088 | 0.106 | 0.644 | 0.382–1.087 | 0.099 |
| Creatinine (mg/dL) | 1.424 | 1.054–1.924 | 0.021 | 1.442 | 1.059–1.964 | 0.02 |
| Antibiotics (Yes/No) | 1.051 | 0.372–2.968 | 0.925 | 0.892 | 0.309–2.574 | 0.833 |
| Sodium (mEq/L) | 1.038 | 0.958–1.124 | 0.363 | - | - | - |
| eGFR(mL/min/1.73m) | 0.989 | 0.973–1.006 | 0.212 | - | - | - |
| Hb (g/dL) | 0.95 | 0.84–1.075 | 0.417 | - | - | - |

HR: hazard ratio, CI: confidence interval, eGFR: estimated glomerular filtration rate, Hb: hemoglobin

## Serum syndecan-1 concentration responded promptly to treatment

Syndecan-1 exists on the surface of several cell types [39–41]. A previous report has revealed that tissue syndecan-1 expression was associated with cardiac fibrosis [42]. However, the present study indicated that syndecan-1 concentration gradually decreased after admission with treatment. If syndecan-1 is the marker of fibrosis, syndecan-1 concentration should not decrease promptly with treatment.

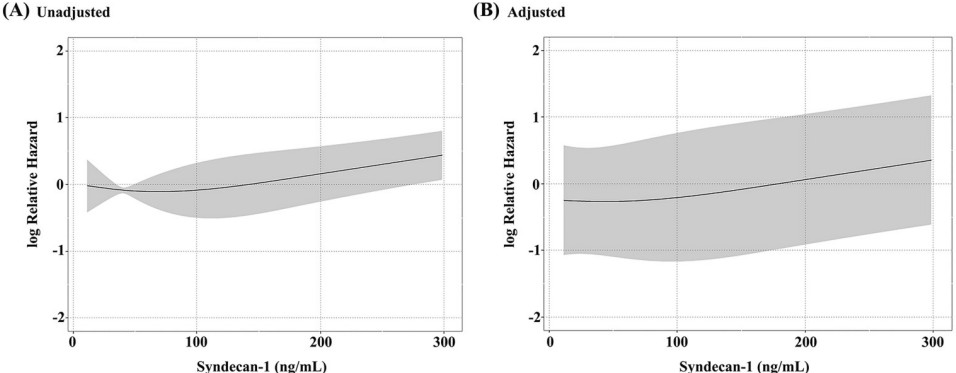

**Fig 2. The relationship between serum syndecan-1 concentration and readmission-free survival.** Univariate and multivariable time-dependent Cox regression model adjusted for age, creatinine (Cre), and antibiotic use. **(A)** The unadjusted time-dependent Cox proportional hazards model. Serum syndecan-1 concentration is associated with readmission-free survival (hazard ratio [HR] for 300 ng/mL of serum syndecan-1 to median, 1.687; 95% confidence interval [CI], 1.1–2.585, p = 0.016). **(B)** The multivariable time-dependent Cox proportional hazards model. Serum syndecan-1 concentration is associated with readmission-free survival even after adjusting for covariates (HR for 300 ng/mL of serum syndecan-1 to median: 1.865, 95% CI: 1.169–2.974, p = 0.009).

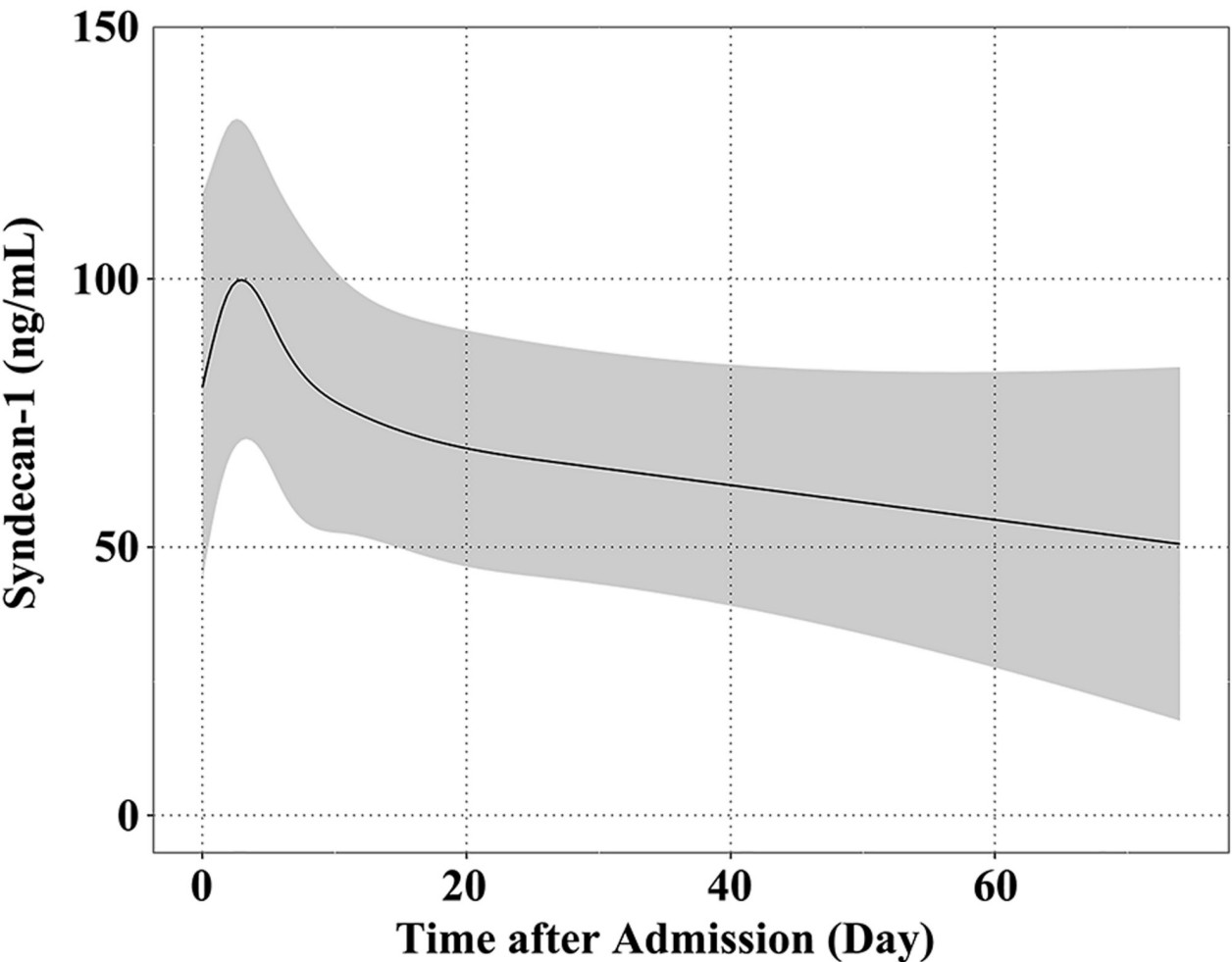

**Fig 3. Time course of serum syndecan-1 concentration after admission.** The concentration of syndecan-1 in serum was 72.4 ng/mL (95% confidence interval [CI], 37.5–107.2) at admission and peaked at 101.21 ng/mL (95% CI, 69.77–132.66) approximately 2.77 days after admission. Subsequently, syndecan-1 concentration decreased gradually.

Conversely, syndecan-1 exists on the surface of systemic vascular endothelial cells and may be released into the serum because of various reasons [41, 43]. Because rapid expansion of plasma volume causes injury to the endothelial glycocalyx [22], syndecan-1, a core protein of the endothelial glycocalyx, may be released into the serum, thereby increasing its concentration in circulation. Moreover, because the turnover of syndecan-1 expression on the pulmonary capillaries is rapid [44], its concentration may sharply increase in the systemic fluid volume. In brief, syndecan-1 may be a perceptive marker for assessing the state of volume overload through the state of heart failure. Achieving readmission-free survival for patients with heart failure with high syndecan-1 concentration was simple in the present study because the systemic fluid volume was abundant, despite the patients having received the treatment.

### Serum syndecan-1 concentration and readmission-free survival

Several factors modified the relationship between readmission-free survival and serum syndecan-1 concentration. Increased syndecan-1 concentration after administration of hANP may indicate unsatisfactory control of congestion. Using antibiotics may pose complications of

**Table 3. Predictors for readmission/death interacting with syndecan-1.**

| Predictor names | p for interaction |
|---|---|
| Beta blockers | 0.885 |
| ACE inhibitor/ARB | 0.661 |
| Statins | 0.495 |
| Antiplatelets | 0.161 |
| Anticoagulants | 0.895 |
| Loop diuretics | 0.248 |
| Spironolactone | 0.021 |
| Tolvaptan | 0.205 |
| Catecholamine | 0.175 |
| Human atrial natriuretic peptide | 0.007 |
| Antibiotics | 0.008 |
| Albumin | 0.476 |
| Aspartate aminotransferase | 0.023 |
| Alanine transaminase | 0.002 |
| Creatinine kinase | <0.001 |
| Triglyceride | 0.995 |
| Total cholesterol | 0.485 |
| High density lipoprotein-cholesterol | 0.576 |
| Low density lipoprotein-cholesterol | 0.605 |
| Blood urea nitrogen | <0.001 |
| Creatinine | 0.016 |
| Sodium | 0.073 |
| Potassium | 0.325 |
| Chlorine | 0.383 |
| C-reactive protein | 0.262 |
| Estimated glomerular filtration rate | 0.244 |
| White blood cells number | 0.12 |
| Hemoglobin concentration | 0.009 |
| Hematocrit | 0.005 |
| Hemoglobin A1c | 0.788 |
| LVDd | 0.841 |
| LVDs | 0.685 |
| IVST | 0.51 |
| PWT | 0.03 |
| LVEF | 0.418 |
| LAD | 0.39 |
| LAVI | 0.026 |
| IVC | 0.284 |

ACE: angiotensin-converting enzyme, ARB: angiotensin II receptor blocker, LVDd: left ventricular diastolic diameter, LVDs: left ventricular systolic diameter, IVST: interventricular septum thickness, PWT: posterior wall thickness, LVEF: left ventricular ejection fraction, LAD: left atrial dimension, LAVI: left atrial volume index, IVC: diameter of the inferior vena cava.

infection such as pneumonia. Moreover, increased ALT and AST concentrations may represent congestive hepatopathy.

Low CK and BUN concentrations also appeared to modify the relationship due to heart failure in patients with low syndecan-1 concentration. These parameters may indicate low

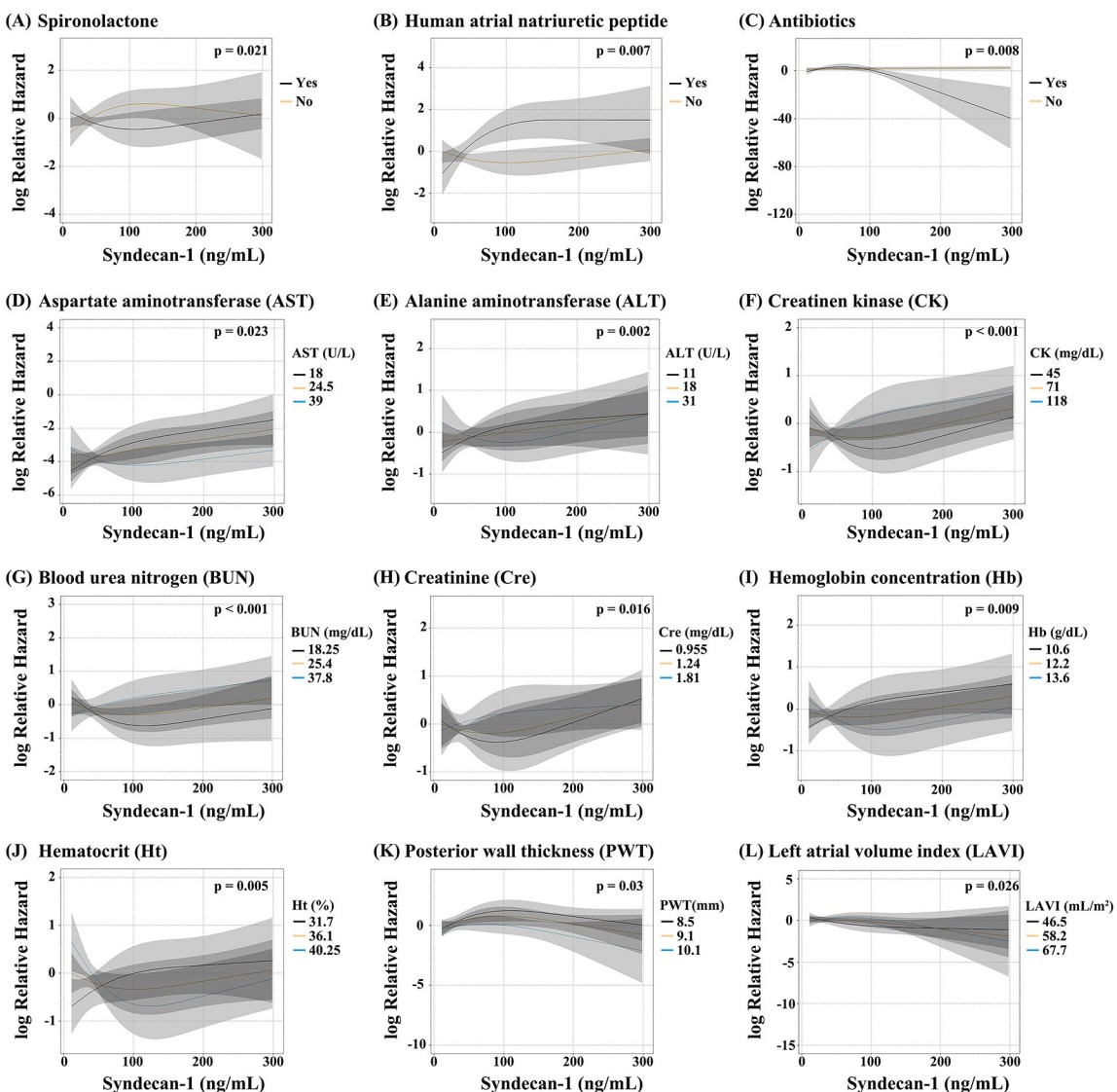

**Fig 4. Interaction effects showing that the parameters modify the relationship between readmission-free survival and serum syndecan-1 concentration.** (**A**) spironolactone, (**B**) human atrial natriuretic peptide, (**C**) antibiotics, (**D**) aspartate aminotransferase (AST), (**E**) alanine aminotransferase (ALT), (**F**) creatine kinase (CK), (**G**) blood urea nitrogen (BUN), (**H**) creatinine (Cre), (**I**) hemoglobin concentration (Hb), (**J**) hematocrit (Ht), (**K**) posterior wall thickness (PWT), and (**L**) left atrial volume index (LAVI). Black, orange, and blue lines indicate the 25th percentile, mean, and 75th percentile of each parameter, respectively.

nutrition status. Patients with heart failure often suffer from muscle atrophy because of chronic low nutrition and frailty [45]. In fact, muscle weakness reportedly affects the prognosis and progression of cardiovascular disease [46]. These results suggest that low nutrition status and excess fluid volume worsen the prognosis of heart failure.

The present study also showed that Hb and Ht modified the relationship between readmission-free survival and serum syndecan-1 concentration. This may be because the hemoconcentration increased wall shear stress and degraded endothelial glycocalyx [47, 48], which were related to syndecan-1 concentrations [33].

Moreover, echocardiography revealed that PWT and LAVI modified the relationship between readmission-free survival and serum syndecan-1 concentration. We believe that these

parameters have a direct relationship with heart failure. Left ventricular hypertrophy has a strong relationship with cardiovascular events such as myocardial infarction and sudden death [49, 50]. Increase in LAVI indicates a complication in atrial fibrillation, which can lead to heart failure and vice versa [51].

Here, we used the time-dependent Cox regression model to account for multiple measurements on a person [52]. This model is an extension of the Cox regression model that only considers risk factors at baseline. Hence, the time-dependent Cox model has the advantage of being able to assess risk posed by syndecan-1 concentration at each measurement point. Specifically, it allows us to assess the association between outcome (readmission or death) at the next measurement and syndecan-1 concentration at the previous measurement at each measurement point.

## Limitations

As a limitation of the present work, the patients included in the study were not classified according to the New York Heart Association functional classification. However, our aim was not to create a diagnostic model but to evaluate the association between syndecan-1 concentration and outcome. In addition, the sample size was small, and data were obtained from a single institution. Therefore, further large scale studies are needed to confirm our findings.

## Conclusions

In conclusion, serum syndecan-1 concentrations, which may indicate injury to the endothelial glycocalyx, can predict readmission-free survival in patients with heart failure. Additionally, the syndecan-1 concentration may be a fluid volume marker for heart failure and may be useful in its management.

## Supporting information

**S1 Fig. Age and sex appeared to modify the relationship due to heart failure.**
(TIF)

**S1 Table. Basal heart disease.**
(DOCX)

**S2 Table. Medication.**
(DOCX)

**S3 Table. The association between BUN/Cre and syndecan-1 concentration.**
(DOCX)

## Acknowledgments

We thank Yasuko Nogaki and Shoko Kumazaki for their technical assistance. We would like to thank Editage (www.editage.com) for English language editing.

## Author Contributions

**Conceptualization:** Yuichiro Kitagawa, Itta Kawamura, Hideshi Okada, Kodai Suzuki, Shozo Yoshida.

**Data curation:** Itta Kawamura, Keiko Suzuki, Akio Suzuki.

**Formal analysis:** Takuma Ishihara, Akio Suzuki.

**Funding acquisition:** Yuichiro Kitagawa, Hideshi Okada, Kodai Suzuki, Tetsuya Fukuta, Ryu Yasuda, Yoshinori Kakino, Takahiro Yoshida, Tomoaki Doi, Shozo Yoshida, Shinji Ogura.

**Investigation:** Chihiro Takada, So Sampei, Soichiro Kano, Kohei Kondo, Hirotaka Asano, Yugo Wakayama, Ryo Kamidani, Yuki Kawasaki, Hirotsugu Fukuda, Ayane Nishio, Takahito Miyake, Tetsuya Fukuta, Ryu Yasuda, Hideaki Oiwa, Yoshinori Kakino, Nagisa Miyazaki, Takatomo Watanabe, Tomoaki Doi, Akio Suzuki.

**Methodology:** Keiko Suzuki, Hiroyuki Tomita, Nagisa Miyazaki.

**Supervision:** Hitoshi Matsuo, Shinji Ogura.

**Writing – original draft:** Yuichiro Kitagawa.

**Writing – review & editing:** Hideshi Okada, Hiroyuki Tomita, Kodai Suzuki, Takahiro Yoshida, Tomoaki Doi, Akio Suzuki, Shozo Yoshida.

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
