## [Decision Letter · Decision Letter 0]

12 Jul 2021

PONE-D-21-12372

Serum syndecan-1 concentration in hospitalized patients with heart failure may predict readmission-free survival

PLOS ONE

Dear Dr. Okada,

Thank you for submitting your manuscript to PLOS ONE. After careful consideration, we feel that it has merit but does not fully meet PLOS ONE’s publication criteria as it currently stands. Therefore, we invite you to submit a revised version of the manuscript that addresses the points raised during the review process.

ACADEMIC EDITOR: All issues raised by expert reviewers are required. In particular, the authors should highlight limitations of the study and include in the title that it is a pilot study. Moreover, the authors should better characterize the clinical, hemodynamics and echocardiographic profile of the patients.

We look forward to receiving your revised manuscript.

Kind regards,

Vincenzo Lionetti, M.D., PhD

Academic Editor

PLOS ONE

Journal Requirements:

Additional Editor Comments (if provided):

Reviewers' comments:

Reviewer's Responses to Questions

**Comments to the Author**

1. Is the manuscript technically sound, and do the data support the conclusions?

Reviewer #1: No

Reviewer #2: Yes

Reviewer #3: Partly

2. Has the statistical analysis been performed appropriately and rigorously? 

Reviewer #1: No

Reviewer #2: Yes

Reviewer #3: Yes

3. Have the authors made all data underlying the findings in their manuscript fully available?

Reviewer #1: No

Reviewer #2: Yes

Reviewer #3: Yes

4. Is the manuscript presented in an intelligible fashion and written in standard English?

Reviewer #1: Yes

Reviewer #2: No

Reviewer #3: Yes

5. Review Comments to the Author

Reviewer #1: the Authors examined if circulating values of syndecan-1, a protein released from the endothelial glicocalix into the bloodstream following endothelial damage, holds prognostic significance in patients with acute heart failure. Although the study is original, the study has important limitations that do not allow a meaningful interpretation of results. First, this is a retrospective study evaluating a small number of patients from a single center. Second, readmission-free survival was defined as the time to readmission for worsened heart failure or death from any cause, which is not acceptable. Third, the model for multivariable adjustment was arbitrarily selected and did not include established outcome predictors such as natriuretic peptides or left ventricular ejection fraction.

Reviewer #2: The paper presents some originality concerning association of syndecan-1 concentration and patients readmission-free survival. Statistical methods are valid, correctly applied, and sufficiently documented to allow replication studies.

If it is possible for authors to explain why did they choose antibiotics for univariate and multivariable time-dependent Cox regression analyses adjustments, instead of heart failure medication? And to outline why did the participants in the study used antibiotics at al, because this kind of therapy may interfere with the results, e.g. syndecan-1 plasma concentration. The authors should also check the abbreviations in the text. I also recommend that data should not be repeated in the tables and in the body text, (e.g. medication, or underlying causes for heart failure). The authors may update the reference list with this relevant publication about syndecan-1 in heart failure patients.

Mitic VT, Stojanovic DR, Deljanin Ilic MZ, Stojanovic MM, Petrovic DB, Ignjatovic AM, Stefanovic NZ, Kocic GM, Bojanic VV.Cardiac Remodeling Biomarkers as Potential Circulating Markers of Left Ventricular Hypertrophy in Heart Failure with Preserved Ejection Fraction.Tohoku J Exp Med. 2020;250(4):233-242

The writing should be improved, and if it is possible for authors, to be softer than it is now. However, the overall quality of the paper is good, and with particular changes it may be considered for publication.

Reviewer #3: The authors presented an elegant study of the role of Sydecan-1 in predicting death and readmission from heart failure. But the authors did not adequately characterize these patients. I think the hemodynamic profile and the echocardiographic characteristics should be included in the article.

Aldo I would like to know why the authors chose values above and below the Sydecan-1 median to determine groups.

6. PLOS authors have the option to publish the peer review history of their article (what does this mean?). If published, this will include your full peer review and any attached files.

Reviewer #1: No

Reviewer #2: No

Reviewer #3: No

---

## [Author Response · Author response to Decision Letter 0]

24 Sep 2021

Responses to the reviewers’ comments

Reviewer #1: 

Comment 1: Although the study is original, the study has important limitations that do not allow a meaningful interpretation of results. First, this is a retrospective study evaluating a small number of patients from a single center. 

Response: Thank you for critically reviewing our manuscript. As pointed out, we have added the aforementioned limitations in the Discussion section of the revised manuscript at the following instance:

Page 23, lines 316–318: 

“In addition, the sample size was small and data were obtained from a single institution. Therefore, further large scale studies are needed to confirm our findings.”

Comment 2: Readmission-free survival was defined as the time to readmission for worsened heart failure or death from any cause, which is not acceptable. 

Response: We apologize for the insufficient information. We have revised the definition of readmission-free survival in the revised manuscript as follows:

Page 9, lines 120–123: 

“The primary outcome of the study was readmission-free survival, defined as the time from the first admission to readmission for worsened heart failure or death from any cause, which was assessed at 30 months after discharge from the hospital.”

Comment 3: The model for multivariable adjustment was arbitrarily selected and did not include established outcome predictors such as natriuretic peptides or left ventricular ejection fraction.

Response: Thank you for a critical comment. Several previous reports have revealed that serum syndecan-1 concentration is influenced by infection [reference numbers 18, 27, 28, 37 in the revised manuscript]. Therefore, to exclude the effect of infection due to endothelial glycocalyx injury, we chose to consider antibiotic use as a confounding factor for univariate and multivariable time-dependent Cox regression analyses. We also needed to avoid overfitting to ensure the reproducibility of our results. Since overfitting can be avoided if the number of events is less than or equal to the number of events divided by 10, it was necessary to limit the number of variables to a factor +3 adjustment variables (Reviewer-only reference).

Furthermore, the reason for choosing age and creatinine as confounding factors is that syndecan-1 concentration in serum is affected by these factors [reference numbers 32–36 in the revised manuscript].　 

Conversely, a previous study has reported that syndecan-1 cannot predict readmission in patients of heart failure with preserved ejection fraction [reference number 30 in the revised manuscript].

Considering the priority of the effect on readmission, only age, creatinine level, and antibiotic use were considered as confounding factors.

We have added this explanation and relevant references in the revised manuscript as follows:

Page 10, lines 133–135: 

“Covariates in the model, including age [32, 33], Cre [33–36], and antibiotic use, indicating the presence of an infectious disease [18, 27, 28, 37], were selected a priori for their clinical relevance.”

Revised references

18. Chelazzi C, Villa G, Mancinelli P, De Gaudio AR, Adembri C. Glycocalyx and sepsis-induced alterations in vascular permeability. Crit Care. 2015;19:26. Epub 2015/04/19. doi: 10.1186/s13054-015-0741-z. PubMed PMID: 25887223; PubMed Central PMCID: PMCPMC4308932.

27. Ostrowski SR, Haase N, Muller RB, Moller MH, Pott FC, Perner A, et al. Association between biomarkers of endothelial injury and hypocoagulability in patients with severe sepsis: a prospective study. Crit Care. 2015;19:191. Epub 2015/04/25. doi: 10.1186/s13054-015-0918-5. PubMed PMID: 25907781; PubMed Central PMCID: PMCPMC4423170.

28. Puskarich MA, Cornelius DC, Tharp J, Nandi U, Jones AE. Plasma syndecan-1 levels identify a cohort of patients with severe sepsis at high risk for intubation after large-volume intravenous fluid resuscitation. J Crit Care. 2016;36:125-9. Epub 2016/11/05. doi: 10.1016/j.jcrc.2016.06.027. PubMed PMID: 27546760; PubMed Central PMCID: PMCPMC6371967.

30. Tromp J, van der Pol A, Klip IT, de Boer RA, Jaarsma T, van Gilst WH, et al. Fibrosis marker syndecan-1 and outcome in patients with heart failure with reduced and preserved ejection fraction. Circ Heart Fail. 2014;7(3):457-62. Epub 2014/03/22. doi: 10.1161/CIRCHEARTFAILURE.113.000846. PubMed PMID: 24647119.

32. Machin DR, Bloom SI, Campbell RA, Phuong TTT, Gates PE, Lesniewski LA, et al. Advanced age results in a diminished endothelial glycocalyx. Am J Physiol Heart Circ Physiol. 2018;315(3):H531-H9. Epub 2018/05/12. doi: 10.1152/ajpheart.00104.2018. PubMed PMID: 29750566; PubMed Central PMCID: PMCPMC6172638.

33. Oda K, Okada H, Suzuki A, Tomita H, Kobayashi R, Sumi K, et al. Factors Enhancing Serum Syndecan-1 Concentrations: A Large-Scale Comprehensive Medical Examination. J Clin Med. 2019;8(9). Epub 2019/08/30. doi: 10.3390/jcm8091320. PubMed PMID: 31462009; PubMed Central PMCID: PMCPMC6780947.

34. Liborio AB, Braz MB, Seguro AC, Meneses GC, Neves FM, Pedrosa DC, et al. Endothelial glycocalyx damage is associated with leptospirosis acute kidney injury. Am J Trop Med Hyg. 2015;92(3):611-6. Epub 2015/01/28. doi: 10.4269/ajtmh.14-0232. PubMed PMID: 25624405; PubMed Central PMCID: PMCPMC4350560.

35. Padberg JS, Wiesinger A, di Marco GS, Reuter S, Grabner A, Kentrup D, et al. Damage of the endothelial glycocalyx in chronic kidney disease. Atherosclerosis. 2014;234(2):335-43. Epub 2014/04/15. doi: 10.1016/j.atherosclerosis.2014.03.016. PubMed PMID: 24727235.

36. Suzuki K, Okada H, Sumi K, Tomita H, Kobayashi R, Ishihara T, et al. Serum syndecan-1 reflects organ dysfunction in critically ill patients. Sci Rep. 2021;11(1):8864. Epub 2021/04/25. doi: 10.1038/s41598-021-88303-7. PubMed PMID: 33893369; PubMed Central PMCID: PMCPMC8065146.

37. Burke-Gaffney A, Evans TW. Lest we forget the endothelial glycocalyx in sepsis. Crit Care. 2012;16(2):121. Epub 2012/04/13. doi: 10.1186/cc11239. PubMed PMID: 22494667; PubMed Central PMCID: PMCPMC3681368.

Reviewer-only reference

Peduzzi P, Concato J, Feinstein A, Holford TR. Importance of events per independent variable in proportional hazards regression analysis. II. Accuracy and precision of regression estimates. J Clin Epidemiol. 1995;48(12):1503-10.

 

Reviewer #2: 

Comment 1: If it is possible for authors to explain why did they choose antibiotics for univariate and multivariable time-dependent Cox regression analyses adjustments, instead of heart failure medication? And to outline why did the participants in the study used antibiotics at all, because this kind of therapy may interfere with the results, e.g., syndecan-1 plasma concentration. 

Response: Thank you for a critical comment. Several previous reports have revealed that serum syndecan-1 concentration is influenced by infection [reference numbers 18, 27, 28, 37 in the revised manuscript]. Therefore, to exclude the effect of infection due to endothelial glycocalyx injury, we chose to consider antibiotic use as a confounding factor for univariate and multivariable time-dependent Cox regression analyses. We also needed to avoid overfitting to ensure the reproducibility of our results. Since overfitting can be avoided if the number of events is less than or equal to the number of events divided by 10, it was necessary to limit the number of variables to a factor +3 adjustment variables (Reviewer-only reference).

Furthermore, the reason for choosing age and creatinine as confounding factors is that syndecan-1 concentration in serum is affected by these factors [reference numbers 32–36 in the revised manuscript].　 

Conversely, a previous study has reported that syndecan-1 cannot predict readmission in patients of heart failure with preserved ejection fraction [reference number 30 in the revised manuscript].

Considering the priority of the effect on readmission, only age, creatinine level, and antibiotic use were considered as confounding factors.

We have added this explanation and relevant references in the revised manuscript as follows:

Page 10, lines 133–135: 

“Covariates in the model, including age [32, 33], Cre [33–36], and antibiotic use, indicating the presence of an infectious disease [18, 27, 28, 37], were selected a priori for their clinical relevance.”

Revised references

18. Chelazzi C, Villa G, Mancinelli P, De Gaudio AR, Adembri C. Glycocalyx and sepsis-induced alterations in vascular permeability. Crit Care. 2015;19:26. Epub 2015/04/19. doi: 10.1186/s13054-015-0741-z. PubMed PMID: 25887223; PubMed Central PMCID: PMCPMC4308932.

27. Ostrowski SR, Haase N, Muller RB, Moller MH, Pott FC, Perner A, et al. Association between biomarkers of endothelial injury and hypocoagulability in patients with severe sepsis: a prospective study. Crit Care. 2015;19:191. Epub 2015/04/25. doi: 10.1186/s13054-015-0918-5. PubMed PMID: 25907781; PubMed Central PMCID: PMCPMC4423170.

28. Puskarich MA, Cornelius DC, Tharp J, Nandi U, Jones AE. Plasma syndecan-1 levels identify a cohort of patients with severe sepsis at high risk for intubation after large-volume intravenous fluid resuscitation. J Crit Care. 2016;36:125-9. Epub 2016/11/05. doi: 10.1016/j.jcrc.2016.06.027. PubMed PMID: 27546760; PubMed Central PMCID: PMCPMC6371967.

32. Machin DR, Bloom SI, Campbell RA, Phuong TTT, Gates PE, Lesniewski LA, et al. Advanced age results in a diminished endothelial glycocalyx. Am J Physiol Heart Circ Physiol. 2018;315(3):H531-H9. Epub 2018/05/12. doi: 10.1152/ajpheart.00104.2018. PubMed PMID: 29750566; PubMed Central PMCID: PMCPMC6172638.

33. Oda K, Okada H, Suzuki A, Tomita H, Kobayashi R, Sumi K, et al. Factors Enhancing Serum Syndecan-1 Concentrations: A Large-Scale Comprehensive Medical Examination. J Clin Med. 2019;8(9). Epub 2019/08/30. doi: 10.3390/jcm8091320. PubMed PMID: 31462009; PubMed Central PMCID: PMCPMC6780947.

34. Liborio AB, Braz MB, Seguro AC, Meneses GC, Neves FM, Pedrosa DC, et al. Endothelial glycocalyx damage is associated with leptospirosis acute kidney injury. Am J Trop Med Hyg. 2015;92(3):611-6. Epub 2015/01/28. doi: 10.4269/ajtmh.14-0232. PubMed PMID: 25624405; PubMed Central PMCID: PMCPMC4350560.

35. Padberg JS, Wiesinger A, di Marco GS, Reuter S, Grabner A, Kentrup D, et al. Damage of the endothelial glycocalyx in chronic kidney disease. Atherosclerosis. 2014;234(2):335-43. Epub 2014/04/15. doi: 10.1016/j.atherosclerosis.2014.03.016. PubMed PMID: 24727235.

36. Suzuki K, Okada H, Sumi K, Tomita H, Kobayashi R, Ishihara T, et al. Serum syndecan-1 reflects organ dysfunction in critically ill patients. Sci Rep. 2021;11(1):8864. Epub 2021/04/25. doi: 10.1038/s41598-021-88303-7. PubMed PMID: 33893369; PubMed Central PMCID: PMCPMC8065146.

37. Burke-Gaffney A, Evans TW. Lest we forget the endothelial glycocalyx in sepsis. Crit Care. 2012;16(2):121. Epub 2012/04/13. doi: 10.1186/cc11239. PubMed PMID: 22494667; PubMed Central PMCID: PMCPMC3681368.

Reviewer-only reference

Peduzzi P, Concato J, Feinstein A, Holford TR. Importance of events per independent variable in proportional hazards regression analysis. II. Accuracy and precision of regression estimates. J Clin Epidemiol. 1995;48(12):1503-10.

Comment 2: The authors should also check the abbreviations in the text. 

Response: Thank you for thoroughly reviewing our manuscript. We have checked all the abbreviations in the text. Moreover, we have defined abbreviations of the following terms in the revised manuscript for improving readability: left ventricular ejection fraction (LVEF),　aspartate aminotransferase (AST), alanine aminotransferase (ALT), creatine kinase (CK), blood urea nitrogen (BUN), creatinine (Cre), hemoglobin (Hb), hematocrit (Ht), posterior wall thickness (PWT), left atrial volume index (LAVI), percutaneous coronary intervention (PCI), coronary artery bypass gr,afting (CABG), angiotensin-converting enzyme (ACE), angiotensin II receptor blocker (ABR), left ventricular diastolic diameter (LVDd), left ventricular systolic diameter (LVDs), interventricular septum thickness (IVST), and hazard ratios (HRs).

Comment 3: I also recommend that data should not be repeated in the tables and in the body text, (e.g., medication, or underlying causes for heart failure). 

Response: Per your suggestion, we have deleted some sentences in the revised manuscript to avoid redundancy.

Comment 4: The authors may update the reference list with this relevant publication about syndecan-1 in heart failure patients. Mitic VT, Stojanovic DR, Deljanin Ilic MZ, Stojanovic MM, Petrovic DB, Ignjatovic AM, Stefanovic NZ, Kocic GM, Bojanic VV. Cardiac Remodeling Biomarkers as Potential Circulating Markers of Left Ventricular Hypertrophy in Heart Failure with Preserved Ejection Fraction.Tohoku J Exp Med. 2020;250(4):233-242

Response: Thank you for the suggestion. We have added this reference as reference number 31 in the revised manuscript. It has been cited at the following instance:

Page 7, lines 89–91: 

“Additionally, it has been suggested that syndecan-1 is associated with left ventricular hypertrophy in heart failure with preserved ejection fraction [31].

New reference

31. Mitic VT, Stojanovic DR, Deljanin Ilic MZ, Stojanovic MM, Petrovic DB, Ignjatovic AM, et al. Cardiac Remodeling Biomarkers as Potential Circulating Markers of Left Ventricular Hypertrophy in Heart Failure with Preserved Ejection Fraction. Tohoku J Exp Med. 2020;250(4):233-42. Epub 2020/04/17. doi: 10.1620/tjem.250.233. PubMed PMID: 32295985.

Comment 5: The writing should be improved, and if it is possible for authors, to be softer than it is now. 

Response: We have got our revised manuscript checked and proofread by a professional English language editing service. We have attached the certificate of editing for your kind perusal.

 

Reviewer #3: 

Comment 1: I think the hemodynamic profile and the echocardiographic characteristics should be included in the article.

Response: Thank you for a critical suggestion. We have included these data in Table 1 in the revised manuscript. The revised table is given below and the additions are shown in red font.

Table 1: Patients’ Demographics

Characteristics N=152

Age (years), median (IQR) 76 (68–85)

Sex (Female/Male), n (%) 58 (38.2) / 94 (61.8)

Follow-up time per patient (days), median (IQR) 23 (11–230)

Number of measurements per patient, median (IQR) 4 (3–6)

Death, n (%) 21 (13.8)

Readmission, n (%) 46 (30.3)

Basal heart disease, n (%) 

 Hypertensive heart disease 26 (17.1)

 Ischemic heart disease (post-PCI) 21 (13.8)

 Ischemic heart disease (post-CABG) 12 (8.9)

 Ischemic heart disease (conservative treatment) 7 (4.6)

 Arrhythmia (tachycardia) 20 (13.2)

 Arrhythmia (bradycardia) 1 (0.7)

 Dilated cardiomyopathy 17 (11.2)

 Hypertrophic cardiomyopathy 5 (3.3)

 Other cardiomyopathy 6 (4.0)

 Aortic valve stenosis (post-operation) 6 (4.0)

 Aortic valve stenosis (conservative treatment) 3 (2.0)

 Aortic valve insufficiency (post-operation) 1 (0.7)

 Aortic valve insufficiency (conservative treatment) 5 (3.3)

 Mitral valve insufficiency (post-operation) 3 (2.0)

 Mitral valve insufficiency (conservative treatment) 8 (5.3)

 Tricuspid valve insufficiency (conservative treatment) 3 (2.0)

 Congestive disease 2 (1.3)

 Other 6 (3.9)

Medication, n (%) 

 Beta blocker 123 (80.9)

 ACE inhibitor/ARB 100 (65.8)

 Statin 61 (40.1)

Antiplatelet 65 (42.8)

Anticoagulant 76 (50.0)

Loop diuretic 136 (89.5)

Spironolactone 100 (65.8)

Tolvaptan 60 (39.5)

Catecholamine 23 (15.1)

Human atrial natriuretic peptide 96 (63.2)

Antibiotic 43 (28.3)

Echocardiographic characteristics, median (IQR) 

LVDd (mm) 53.2 (46.0–58.1)

LVDs (mm) 41.4 (30.4–51.0)

IVST (mm) 9.0 (8.3–9.7)

PWT (mm) 9.3 (8.4–10.2)

LVEF (%) 40.5 (25.6–59.6)

LAD (mm) 46.7 (43.7–52.0)

LAVI 60.2 (47.4–67.7)

IVC (mm) 8.6 (6.4–12.1)

IQR: interquartile range, PCI: percutaneous coronary intervention, CABG: coronary artery bypass grafting, ACE: angiotensin-converting enzyme, ARB: angiotensin II receptor blocker, LVDd: left ventricular diastolic diameter, LVDs: left ventricular systolic diameter, IVST: interventricular septum thickness, PWT: posterior wall thickness, LVEF: left ventricular ejection fraction, LAD: left atrial dimension, LAVI: left atrial volume index, IVC: diameter of the inferior vena cava.

Comment 2: I would like to know why the authors chose values above and below the Sydecan-1 median to determine groups.

Response: There are three reasons for dividing patients into high and low syndecan-1 concentration groups stratified by the median value. First, a standard criterion has not yet been established for dividing patients based on syndecan-1 concentration in serum. Second, because syndecan-1 has a skewed distribution, it was necessary to use a robust index that is not affected by outliers. Third, stratification by the median value gives the highest statistical power because the sample size in each group would be equal.

---

## [Decision Letter · Decision Letter 1]

9 Nov 2021

Serum syndecan-1 concentration in hospitalized patients with heart failure may predict readmission-free survival

PONE-D-21-12372R1

Dear Dr. Okada,

We’re pleased to inform you that your manuscript has been judged scientifically suitable for publication and will be formally accepted for publication once it meets all outstanding technical requirements.

Kind regards,

Vincenzo Lionetti, M.D., PhD

Academic Editor

PLOS ONE

Additional Editor Comments (optional):

Reviewers' comments:

Reviewer's Responses to Questions

**Comments to the Author**

1. If the authors have adequately addressed your comments raised in a previous round of review and you feel that this manuscript is now acceptable for publication, you may indicate that here to bypass the “Comments to the Author” section, enter your conflict of interest statement in the “Confidential to Editor” section, and submit your "Accept" recommendation.

Reviewer #1: All comments have been addressed

Reviewer #2: All comments have been addressed

Reviewer #3: All comments have been addressed

2. Is the manuscript technically sound, and do the data support the conclusions?

Reviewer #1: Yes

Reviewer #2: Yes

Reviewer #3: Yes

3. Has the statistical analysis been performed appropriately and rigorously? 

Reviewer #1: Yes

Reviewer #2: Yes

Reviewer #3: Yes

4. Have the authors made all data underlying the findings in their manuscript fully available?

Reviewer #1: Yes

Reviewer #2: Yes

Reviewer #3: Yes

5. Is the manuscript presented in an intelligible fashion and written in standard English?

Reviewer #1: Yes

Reviewer #2: Yes

Reviewer #3: Yes

6. Review Comments to the Author

Reviewer #1: The Authors have modified their manuscript according to my comments and suggestions. I have no further comments.

Reviewer #2: (No Response)

Reviewer #3: This is study is a retrospective observational study evaluating the relation between serum syndecan-1 concentration and readmission-free survival of hospitalized patients with decompensated heart failure. Patients had their blood sample collected on admission for measuring syndecan-1 and an echocardiography was performed also on admission. The outcome was the number of days since first admission until readmission or death from any cause, evaluated retrospectively after 30 months. The authors found serum syndecan-1 may predict readmission-free survival in patients with heart failure.

In order to minimize influence of other factors, the authors performed univariate and multivariable analysis. They considered unexpected variables to the reviewers, and did not analyzed variables such as natriuretic peptides, ejection fraction and heart failure medications. After better explanation, we understand the authors selected variables that directly influence levels of syndecan-1, such as infection, renal function and age for the analysis, and they provided references to support their explanation.

The authors have improved the manuscript making it more readable and allowing the reader to interpretate the results considering all the methodologic limitations.

7. PLOS authors have the option to publish the peer review history of their article (what does this mean?). If published, this will include your full peer review and any attached files.

Reviewer #1: No

Reviewer #2: No

Reviewer #3: No

---

## [Editor Report · Acceptance letter]

29 Nov 2021

PONE-D-21-12372R1 

Serum syndecan-1 concentration in hospitalized patients with heart failure may predict readmission-free survival 

Dear Dr. Okada:

I'm pleased to inform you that your manuscript has been deemed suitable for publication in PLOS ONE. Congratulations! Your manuscript is now with our production department. 

Kind regards, 

on behalf of

Prof. Vincenzo Lionetti 

Academic Editor

PLOS ONE